# Variations in Bovine Milk Proteins and Processing Conditions and Their Effect on Protein Digestibility in Humans: A Review of *In Vivo* and *In Vitro* Studies

**DOI:** 10.3390/foods13223683

**Published:** 2024-11-19

**Authors:** Conor J. Fitzpatrick, Daniela Freitas, Tom F. O’Callaghan, James A. O’Mahony, André Brodkorb

**Affiliations:** 1Teagasc, Moorepark Research Centre, Fermoy, Co., P61 C996 Cork, Ireland; conor.fitzpatrick@teagasc.ie (C.J.F.);; 2School of Food and Nutritional Sciences, University College Cork, T12 K8AF Cork, Ireland; 3Vistamilk SFI Research Centre, Teagasc, Moorepark, Fermoy, Co., P61 C996 Cork, Ireland

**Keywords:** caseins, whey proteins, protein polymorphisms, pasteurisation, digestion

## Abstract

Bovine milk proteins account for 10% of the global protein supply, which justifies the importance of thoroughly understanding their digestive processes. Extensive research on digestion is being conducted both *in vivo* and *in vitro*. However, interpretations and comparisons across different studies require a thorough understanding of the methodologies used. Both the rate and extent of milk protein digestion can be affected by several intrinsic and extrinsic factors with potential implications for overall digestibility and physiological responses. Among intrinsic factors, the impact of genetic variants in native milk proteins has emerged as a growing research area. To these, further complexity is added by the processing conditions frequently applied to milk prior to consumption. The main aim of this work is to provide an overview of the current knowledge on the impact of variations in milk protein profiles (particularly whey: casein ratio and protein polymorphisms), the treatments applied during processing (pasteurisation, homogenisation) and consumption (temperature changes) on protein digestion. To support the interpretation of the current literature, this manuscript also presents a historical perspective into research in this field and summarizes the protocols that are most frequently used, presently, on *in vitro* digestion studies.

## 1. Bovine Milk as a Human Food Source

“Milk is food. Alone and unassisted it is capable not only of sustaining life for an indefinite period, but it furnishes all the elements for the complete construction of the human frame; on it alone the infant learns to talk and walk, and develops all the tissues of the system. Adults have lived on it alone for weeks or months, and by it convalescents from grave sicknesses have recovered vitality and strength” [1].

As highlighted by BULKLEY [1], the importance of bovine milk to human nutrition has long been understood. Studies from the late 19th and early 20th century provided observations on the changes that occur to milk in the stomach. As analytical techniques have become more sophisticated, the biochemical mechanisms have been increasingly understood and described. Advances in various technologies can offer potential for greater understanding of the nutritional and biological value of milk proteins.

Bovine milk contains approximately 29–34 g/L of high quality, digestible protein, meaning that consuming 0.5 L can account for 30–40% of a person’s daily protein needs [2,3,4]. Bovine milk provides all of the essential amino acids required for human health. This is especially important in infant nutrition. While breastfeeding is recommended for infants when possible, formula products are commonly used as substitutes, typically formulated from blends of bovine milk proteins, including whey and casein-rich components, to produce an amino acid profile closer to that of human milk [5]. As well as providing amino acids that are needed for human health, certain milk proteins also play a role in transporting minerals needed for human health. For example, α-La is a metalloprotein, binding one Ca^2+^ per mol, which facilitates the passive absorption of Ca^2+^ [6,7].

Studying the digestion of milk proteins is important, as changes to the rates and extents of digestion can impact physiological responses such as gastric emptying dynamics, which can affect satiety and postprandial aminoacidemia (concentration of amino acids in the blood). Techniques such as Magnetic Resonance Imaging (MRI) are increasingly being used as non-invasive methods to study digestion *in vivo* [8]. However, the utilisation of human subjects poses several challenges and limitations, such as ethical considerations, reproducibility, cost and low throughput. Despite these limitations, *in vivo* studies are required for measuring certain physiological outcomes such as the rate of gastric emptying and the rate of aminoacidemia, which is important due to the role of amino acids in muscle protein synthesis. For instance, consuming free amino acids in a readily absorbable state increases the concentration of plasma amino acids compared with intact milk proteins, leading to a higher release of phenylalanine, a precursor to tyrosine, which is vital for neurotransmitter formation; however, both intact and free amino acids have been shown to increase rates of muscle protein synthesis [9]. *In vivo* studies have also shown differences in the rate of aminoacidemia in whey protein vs. caseins; using intrinsically labelled leucine, it has been shown that whey proteins cause transient spikes in aminoacidemia, while casein results in a prolonged release over a longer period [10].

Due to the limitations of *in vivo* studies, *in vitro* digestion systems are commonly used. The increasing sophistication and standardisation of *in vitro* digestion systems, such as the standardised INFOGEST method, has led to an increase in reproducible *in vitro* digestion of bovine milk proteins, without the drawbacks of ethical considerations or low throughput seen in *in vivo* studies [11]. These *in vitro* digestion systems can range from static *in vitro* digestions, where all food, enzymes and fluids are added to a single tube, to dynamic *in vitro* digestions that incorporate complex churning, pH dynamics, temperature control and gastric-emptying mechanics [11,12].

However, while *in vitro* digestion studies can suggest potential physiological outcomes, they cannot completely replicate complex *in vivo* conditions. For these reasons, it is important to consider the benefits and limitations of both *in vivo* and *in vitro* digestion methods when trying to understand the digestion of bovine milk proteins.

This review paper focuses on intrinsic and extrinsic factors affecting the digestion of milk proteins, as well as discussing the various methods used to simulate milk protein digestion. The intrinsic factors reviewed here include whey protein to casein ratio and milk protein polymorphisms. The extrinsic factors reviewed here include the processing of milk and milk consumption temperatures. Certainly, other factors can affect the digestion of milk proteins, and have been reviewed extensively elsewhere. For instance, the effect of physiochemical modifications of milk proteins has been extensively reviewed by Dupont and Tomé [13]. Interestingly, the effect of species on milk protein digestion has been investigated; however, while this matter is touched on briefly in this review, it is extensively reviewed elsewhere [14]. The effect of processing on milk protein digestion has also been reviewed; however, the present study introduces new studies to the discussion [15]. This review will offer several benefits, bringing together both historical and modern research on the digestion of milk proteins, and will serve as a reference point for future research in this area.

### 1.1. Early Observations of Milk Protein Digestion

Studies from the 19th and 20th centuries provide findings on the digestion of milk *in vivo*, which would be difficult to obtain with modern ethical standards. One of the earliest recorded cases of the study of the gastric digestion of milk was carried out on the infamous patient, Alexis St. Martin, who had an external opening into his stomach due to a gunshot wound. These experiments revealed that milk was easily digested and provided the first evidence that heating milk affects its digestion, with raw milk “chymifying” more slowly than boiled milk [16].

Further studies examined the difference in coagulation properties between raw and boiled milk [17]. By feeding raw and boiled milk to a subject at different times, and subsequently having that subject return the samples from the stomach through emesis, a clear picture of what occurred during the gastric digestion of milk was observed. When boiled milk was returned after 30 min, it had formed small, soft, ‘custard-like’ curds, which contrasted to raw milk, which had formed firm, rubbery curds. After 3 h of digestion, raw milk curds were fewer, but larger and firmer with rounded corners. Moreover, it was found that with increasing time, the curds became more and more difficult to regurgitate, requiring water to be consumed in order to evacuate from the stomach. As well as this, the contents became progressively more sour and acidic as time after consumption increased. This indicates that the pH was decreasing as digestion continued due to the secretion of hydrochloric acid from parietal cells in the gastric wall [18].

Mortenson, *et al*. [19] showed similar finding in calves: boiled milk was found to break up more easily and was therefore emptied more rapidly than raw milk using a gastric fistula. As well as this, it was found that raw milk coagulated within 10 min, while boiled milk coagulated within 8–15 min.

These studies provide valuable information on the digestion of milk in humans *in vivo*. However, the author does not condone the methods used to obtain these results due to basic ethical considerations. Certainly, the authors were conflicted on whether or not to discuss these studies due to the use of Alexis St. Martin as a domestic slave. However, the authors concluded that by recognising the contribution of Alexis St. Martin to the field of Gastroenterology, a recognition that he never received during the experiments, that discussing these results is justified.

### 1.2. Modern Explanation of Milk Protein Digestion

Early studies on the digestion of milk proteins provide valuable observations that were later explained using advanced analytical techniques. We now know that the digestion of milk proteins in humans begins in the stomach, where it is exposed to low pH and hydrolytic enzymes such as pepsin. In milk, caseins are organised into micelles, which are complex structures formed from several types of casein proteins (αs1- and S2-, β-casein (β-casein) and κ-casein (κ-casein), as well as salts, mainly calcium phosphate, which is present as colloidal calcium phosphate, and water [20]. Casein micelles stay in solution due to κ-casein proteins protruding from the surface, providing a negative charge and steric repulsion between micelles. However, during digestion, these κ-casein protrusions are cleaved by the action of pepsin, and, if pepsin is not present, the acidic conditions of the stomach will eventually cause the proteins to reach their isoelectric point, thereby eliminating the steric repulsion between micellar surfaces, resulting in coagulation [21]. This is the biochemical explanation of the observations of curd formation in the gastric phase provided in early studies discussed above [17,22]. The characteristics of the coagulum can be influenced by several factors and can lead to differences in rates of protein hydrolysis as shown in an *in vitro* digestion study by Mulet-Cabero, *et al*. [23]. As observed in early studies, a firmer, semi-solid ‘coagulum’ matrix, as seen in raw milk, makes it difficult for pepsin to reach the proteins entrapped within the matrix, while a more porous matrix, as seen in heat-treated milk, gives pepsin easier access, potentially modulating the rate of both protein hydrolysis and gastric emptying [17].

The presence of casein micelles in milk is an important factor in gastric coagulation. In skim milk powder with intact casein micelles, coagulation occurs within 10 min of gastric digestion. However, when casein micelles are disrupted by the removal of calcium, which may be enacted to increase the solubility of casein, coagulation does not occur until after 40 min of gastric digestion [24,25]. Reduced calcium milk protein concentrate can be produced by injecting CO_2_ followed by ultrafiltration [26]. Further, when the amount of micellar calcium phosphate is increased, the rate of digestion of coagulum has been shown to decrease due to stronger coagulum formation [27]. This may be of interest if attempting to modulate the gastric emptying rate, and thereby the rate of protein absorption into the bloodstream.

In contrast to casein, whey protein (WP) does not coagulate in the gastric phase [24]. Native WPs resist gastric digestion due to their globular structure. β-Lg for instance is largely broken down only when it moves from the acidic gastric phase to the neutral duodenal phase [28]. An *in vitro* study using the INFOGEST method has demonstrated that large quantities of intact β-LG were observed following gastric digestion and also showed a rapid disappearance of β-LG at the beginning of the intestinal phase, therefore showing that β-LG is highly susceptible to intestinal digestion [29]. The resistance of native β-LG to gastric digestion is further supported in work by Dupont, *et al*. [30] where it was demonstrated to be highly resistant to adult and infant digestion *in vitro*; however, the lack of protective phosphatidylcholine in the infant stomach makes β-LG slightly more susceptible to acidic and enzymatic degradation [30]. Phosphatidylcholine is an abundant gastric surfactant. Proteins such as α-LA have been shown to be embedded within phosphatidylcholine during gastric digestion, which protects it from digestion [31]. In the small intestine, milk proteins are further hydrolysed by the action of pancreatic enzymes, which include trypsin, chymotrypsin, and carboxypeptidases [13]. Other factors can increase the digestion of proteins, for instance, bile salts have been shown to denature proteins, increasing the rate of hydrolysis [32]. This is particularly interesting in the case of β-LG due to its resistance to digestion, with bile salts being shown to significantly increase the hydrolysis of this protein.

Much work in the area of digestion and human nutrition has been carried out on the digestion of β-LG, largely due to its implications in milk allergy and its potentially hypotensive effects. In terms of allergenicity, β-LG is a major milk protein responsible for milk allergy [33]. However, β-LG from different species does not react in the same way to bovine β-LG [34]. The major differences in bovine vs. caprine β-LG are that caprine β-LG precipitates at pH 6.7 to 7.1 over 80 °C while bovine β-LG does not, and that bovine β-LG has a lower dissociation constant than caprine β-LG, meaning that bovine β-LG remains in dimer format at pH 2.5, while caprine β-LG dissociates to monomers [35]. This difference may allow greater digestion of caprine β-LG compared with bovine; however, more research is required. Through genetic modification, when the free cysteine residue of bovine β-LG, located in the calyx or β-barrel region of the protein at position 121, was substituted with a serine residue, it was found that the amount of this protein that remains intact after gastric hydrolysis decreases from approximately 80 to 35%. Jayat, *et al*. [36] suggested that this increase in hydrolysis is due to increasing accessibility of the enzymes to the binding site due to an increased mobility of the protein.

## 2. *In Vitro* Digestion Systems: Considerations for the Study of Milk Protein Digestion

The goal of *in vitro* digestion system is to mimic the *in vivo* human digestive system. Several systems have been developed to attempt this, with examples of these methods summarised in Table 1. Each of these systems has its specific advantages and limitations which may have implications on the results obtained and conclusions that can be drawn from each study. An understanding of these aspects is essential to ensure the correct study design and outcome interpretation.

Prior to the implementation of a standardised *in vitro* digestion system, several authors had published work, each with different protocols, making comparisons between studies difficult or, in some cases, impossible [39]. In 2014, Minekus, *et al*. [40] proposed a standardised INFOGEST static *in vitro* digestion protocol, which was further optimised in 2019 (INFOGEST 2.0) [11]. The term static is used here as the dynamic aspects of digestion (e.g., flow of secretions, progressive gastric acidification and emptying) are not replicated. Instead, at the start of each digestive phase, food/chyme, enzymes and simulated digestive fluids are all added together, and the pH is adjusted to a fixed value. Typically, three pH adjustments are performed: once at the start of the oral phase (pH 7), once at the start of the gastric phase (pH 3) and again at the start of the intestinal phase (pH 7). As well as this, the duration of both the gastric and intestinal phases is set at 2 h for each, and does not take into account the volume/amount or caloric content of the meal. Because of their simplified approach, static *in vitro* digestion protocols are considered a useful, straightforward tool that facilitates a high throughput. Moreover, for the study of protein digestion, they can be particularly useful for ranking different protein sources according to their digestibility and possibly to investigate the release of bioactive sequences in the gastrointestinal tract [41]. For milk proteins in particular, the gastric and intestinal endpoints of INFOGEST’s static digestion protocol showed a good approximation to the pig *in vivo* digestion [42]. This protocol is inherently limited in its capacity to mimic the dynamic evolution of protein hydrolysis that occurs *in vivo*; however, it has been successfully validated with dairy products for gastric and intestinal endpoints [38,43,44,45]. Another important aspect to consider is that this protocol is only appropriate for simulating adult digestions. However, since its publication, several papers have examined the need for adjustments when studying different population groups according to, for example, sex [46], disease or age, e.g., infants [47] and older adults [48].

At present, an international consensus has been reached by the INFOGEST network, as part of an EU project Eat4Age, on the appropriate adaptations of a static *in vitro* model to reflect digestion in older adults [48]. Several differences to the adult protocol are recommended. More specifically, in the oral phase, the elderly protocol outlines a more specific protocol for simulating chewing. However, the food to oral fluid ratio remains the same in both protocols, as does the time, pH and enzyme concentration (1:1 ratio, 2 min, pH 7, 75 U of salivary amylase/mL). The major differences occur in the gastric and intestinal phases. In the gastric phase of the adult model, the pH is set to 3, with a pepsin and lipase concentration of 2000 and 60 U/mL of gastric contents, respectively, and the time of gastric digestion is 2 h. In the elderly protocol, the pH is set to 3.7, and reduced pepsin and lipase concentrations are used (1200 and 26 U/mL of gastric contents, respectively). As well as this, the gastric digestion time is increased to 3 h. In the intestinal phase, the pH and time remain the same as the adult; however, the pancreatin and bile salts concentrations are both decreased. A recent *in vitro* digestion study has shown that gastric digestion of proteins in dairy products is significantly hindered in elderly models compared with the adult. However, the digestion of these proteins in the intestinal phase is not affected [49].

Work has also been carried out to develop an infant static *in vitro* digestion model. Menard and collaborators carried out research leading to a ‘first step towards a consensus’ method in 2018 [47]. This model represents a method to simulate digestion in a full-term new-born at 28 days of life. In this model, due to an observed low retention time in the oral phase *in vivo*, no oral phase is carried out. An interesting difference in this protocol compared with both the adult and elderly models is a different ratio of food to gastric fluid (63:37 vs. 1:1 in the infant vs. adult, respectively). As well as this, the enzyme concentration is dramatically decreased, with 268 and 19 U/mL of gastric fluid for pepsin and lipase, respectively. The pH is also increased to pH 5.3, and the gastric digestion time is reduced to 1 h from 2 h in the adult model. In the intestinal phase, a difference in digestive enzymes is again recommended, with the infant model using a pancreatin concentration of 90 U of lipase activity/mL, while the adult model recommends 100 U of trypsin activity/mL. Moreover, a reduction in bile salt concentrations from 10 to 3.1 mmol/L is recommended, as well as a decrease in pH to 6.6. When the adult and infant models were compared in terms of milk protein digestion in the gastric phase, it was found that all caseins were hydrolysed significantly more extensively and rapidly in the adult model compared with the infant model; however, these differences were eliminated in the intestinal phase, with all proteins remaining after 5 min [47]. In terms of lipolysis, no difference was seen between models in the gastric phase; however, due to decreased quantities of surfactants and lipases in the infant compared with the adult, lower levels of lipolysis were observed in the infant vs. adult. Adult gastric conditions also differ to infant gastric conditions in terms of phosphatidylcholine content. Phosphatidylcholine is a lipid secreted from the gastric mucosa and is present in milk. This lipid has been shown to act as a surfactant, with a protective effect on β-Lg. In a study by Mandalari, *et al*. [50], the breakdown of β-Lg was compared in simulated adult vs. infant digestion. In the infant model, a lower concentration of phosphatidylcholine was used compared with adult digestion. No effect on gastric digestion of β-Lg was found between infant and adult models; however, after 30 min of duodenal digestion, β-Lg degraded to a lesser extent when phosphatidylcholine was present [30]. This suggests that this lipid binds to the β-Lg, potentially blocking pepsin through steric repulsion.

The INFOGEST research network has also developed an *in vitro* semi-dynamic digestion protocol based on the static adult model [37]. In this model, the oral and intestinal phases remain ‘static’, while the gastric phase incorporates dynamic elements, including constant stirring using an overhead stirrer, the continuous addition of simulated gastric fluid and enzymes, a gradually decreasing pH (finish at pH 2) and gastric emptying based on an emptying rate of 2 kcal/min [37].

As well as *in vitro* models to replicate different gastrointestinal conditions due to age, a first attempt on a gender-specific semi-dynamic *in vitro* digestion method has been published recently [46]. Based on *in vivo* data, Lajterer, *et al*. [46] altered gastrointestinal conditions dependent upon sex. For example, a lower concentration of pepsin was used for females compared with males in the gastric phase, as well as a longer intestinal phase (3 vs. 2 h) along with lower chymotrypsin concentrations. While only small differences in the digestion of milk proteins was observed between male and female conditions, it is important to consider these factors going forward to ensure an accurate representation of *in vivo* digestion.

To enhance the similarity between *in vitro* systems and the *in vivo* situation, especially in terms of digestion kinetics, several dynamic models have been published. These models can be split into two types, mono-compartmental and multi-compartmental. Mono-compartmental systems simulate one part of the human digestion system, while multi-compartmental systems simulate several parts. Examples of mono-compartmental systems are the Human Gastric Simulator (University of California, Davis) and the Dynamic Gastric Model (DGM) (Institute of Food Research, Norwich, UK) [51,52]. These systems are similar in that they allow the precise control of pH, temperature, gastric secretions and gastric emptying, as well as the ability to mimic both chemical and physical breakdown of food as seen *in vivo*. Several key differences exist between the two systems. In the DGM, gastric mixing is achieved through contractions that are controlled by water pressure using a piston and barrel system, while the HGS uses rollers along the outside of the artificial stomach. The DGM mimics *in vivo* digestion at the end of the gastric phase by also mimicking the ‘housekeeper wave’ to remove all contents from the stomach, which does not seem to occur in the HGS.

Even more complex are the multi-compartmental models, which incorporate dynamic systems for several stages of digestions. These include the DIDGI system (INRAE, France), the SIMulator of Gastrointestinal tract (SIMGI) (Food Science Research CIAL, CSIC-UAM, Madrid, Spain), Engineered Stomach and Small Intestinal (ESIN) (University of Auvergne, Clermont-Ferrand, France), TIM and Tiny TIM (TNO, the Netherlands) and SHIME (Ghent University, Belgium) [53]. The DIDGI system (Figure 1) simulates digestion in the gastric and intestinal phase. All functions, such as meal flow, digestive fluid flow and enzyme flow, are controlled by a computer program (SToRM^®^) (INRA, Grignon, France) [54]. The gastric and intestinal phases are linked by a Teflon membrane. Unlike ARCOL, anaerobic conditions in the small intestine are simulated by pumping in nitrogen gas. This model is limited by basic mixing function not simulating the *in vivo* situation. While these models are highly useful, they do not mimic gastric motility such as is seen in the *in vivo* situation due to peristalsis and segmentation actions of the muscles of the gastric wall which may impact the digestion of protein [55].

The TIM system is a still more advanced system, incorporating the stomach, duodenum, jejunum and ileum, and mimicking *in vivo* mixing conditions using peristaltic waves through water pressure on the outside of the flexible walls [56]. Most significantly, the TIM-2 system has the ability to mimic intestinal absorption in the jejunum and ileum using a combination of dialysis and filtration, as well as replicating the microbial flora using human or animal faeces. The ESIN system has the ability to more accurately mimic the condition of food when it enters the gastric phase compared with all other systems [57]. This is due to the use of a reservoir that stores food and incorporates a salivary ampoule that mixes food with saliva. These two features combined, allow for the progressive release of food particles into the gastric phase, which have similar characteristics to the *in vivo* situation. The SIMGI system, similar to the TIM system, uses water pressure to simulate peristaltic movement [58]. A distinctive feature of the SIMGI is the ability to mimic physiological conditions and disease states in the gastrointestinal system due to its automated working parameters. All of the presented methods have been extensively discussed previously in Dupont, *et al*. [59]. Dynamic digestion models are useful for studying complex physiology that cannot be achieved with semi-dynamic or static systems. For example, TIM-1 has been used to show that β-LG and glycomacropeptide are not only the most resistant milk proteins to gastric digestion but also that they are emptied last from the gastric to the intestinal phase [60,61].

Several studies have compared the previously discussed *in vitro* digestion methods to each other, and to the *in vivo* situation in terms of bovine milk protein digestion. For instance, Egger, *et al*. [44] compared the digestion of milk proteins using the *in vitro* static digestion approach published by Minekus, *et al*. [40] with *in vivo* digestion in pigs. Considering that in both the *in vitro* digestion and the *in vivo* digestion, no intact casein and a small amount of intact β-LG was seen in the duodenal phase, it was concluded that the *in vitro* system was successful at mimicking the *in vivo* situation. This was further confirmed by similar peptide patterns at the end of digestion using mass spectrometry. Following the publication of a standardised static *in vitro* digestion method, Egger, *et al*. [44] again examined the similarity between the new static *in vitro* digestion system and *in vivo* digestion in pigs, as well as including an *in vitro* dynamic digestion (DIDGI system) in the comparison [11]. The *in vitro* static and dynamic digestions differed to the *in vivo* gastric phase in terms of casein digestion. In both *in vitro* studies, casein disappeared during the gastric phase, while in the *in vivo* situation, gastric contents were split into a solid and liquid phase, where all casein had disappeared in the liquid phase, and intact casein remained in the solid phase. However, this study did not use the same protein concentrations in skim milk powders, with the *in vivo* study using a 33% protein concentration, compared with just the 10% used in the *in vitro* studies resulting in a weaker coagulum *in vitro*. In terms of the release of free amino acids during digestion, it was found that *in vitro* dynamic digestion more closely matched the *in vivo* situation. This may be due to several factors, including the dynamic system incorporating mixing dynamics more closely related to the *in vivo* situation, and a pH curve more accurately matching the *in vivo* situation, which can affect the activity of digestive enzymes such as pepsin and lipase. The SIMGI system has also been validated to the *in vivo* situation when digesting WP. By using peptide profile analysis, Barbé, *et al*. [62] found that after the duodenal phase, peptide fragments resistant to digestion were very similar in both the *in vivo* situation (using mini-pigs) and in the SIMGI *in vitro* dynamic digestion system, indicating that this is an accurate model. The correlation between results in *in vitro* and *in vivo* milk protein digestion studies have been critically reviewed previously, concluding that both gastric and intestinal endpoints of digestion *in vitro* and *in vivo* were comparable [41].

### Digesta Analysis and Characterisation Techniques

During *in vitro* digestions, samples are taken at set time points during oral, gastric and intestinal phases. These samples, or digesta, can be analysed in several ways to understand the rate and extent of digestion. These techniques range from qualitative techniques to complex quantitative techniques. To study phenomena such as aggregation and coagulation, microscopy techniques can be useful. Common microscopy techniques used to analyse *in vitro* digestions include light microscopy, confocal laser scanning microscopy (CLSM) and scanning electron microscopy (SEM) [24]. Other visualisation techniques, such as MRI, can be powerful tools to examine particle formation, disintegration and particle volume, as shown by Fitzpatrick, *et al*. [63].

Common lab assays can also be conducted to quantify the extent of digestion. These include the ortho-phthalaldehyde (OPA) assay to measure the amount of free amines in a sample as an indicator of protein hydrolysis, the bicinchoninic acid (BCA) assay to quantify amount of protein in a sample, or the SDS-PAGE technique to visualise the breakdown of large proteins into smaller fragments [64,65,66].

More sophisticated techniques can also be used to perform more advanced separation and quantification. These include chromatography methods such as size exclusion chromatography (SEC-HPLC) [67]. These techniques can be useful to analyse peptide release and monitor the breakdown of specific proteins. Even more advanced is mass spectrometry (MS), which can be used in tandem as MS/MS to obtain individual peptides and amino acid sequences [38]. Each of these techniques has benefits and drawbacks, with the simpler techniques offering high throughput at a low cost; however, the more advanced techniques offer more complex insights.

These techniques can also be used to analyse *in vivo* digesta samples, and can be taken from human trials, or from pig trials simulating human digestion. However, techniques such as the Digestible Indispensable Amino Acid Score (DIAAS) and the Protein Digestibility-Corrected Amino Acid Score (PDCAAS) can only be carried out *in vivo* and are used to measure protein quality based on amino acid absorption [68,69]. Both industry and the scientific community are asking for *in vitro* alternatives to estimate the *in vitro* DIAAS of existing and new alternative proteins. The static INFOGEST method is currently being considered as an *in vitro* protein digestibility ISO standard method [42,43].

## 3. Variations in the Profiles of Milk Proteins

### 3.1. Whey Protein to Casein Ratio

Variations in the ratio of bovine milk casein to WP content, which is approximately 80:20, can be naturally influenced by breed and somatic cell count (SSC), or artificially altered for different dairy products. Studies show that different casein:WP ratios alter amino acid appearance rates in the blood post-ingestion [10]. Casein-rich meals form a firm curd in the stomach, slowing gastric emptying compared with whey-rich meals, which form a transient curd that quickly returns to a liquid phase. This difference in curd formation affects gastric emptying rates of protein from the stomach, with proteins that form firm curds being emptied more slowly than proteins that remain in the liquid phase or form soft curds. An *in vitro* digestion study by Mulet-Cabero, *et al*. [70] showed that varied casein:whey ratios (0:100, 20:80, 50:50, 80:20) form a curd after 10 min, with low-casein samples solubilizing quickly and high-casein samples remaining firm. This study also suggested an enzymatic effect on coagulation, likely through κ-casein cleavage at Phe105-Met106, reducing steric repulsion and allowing micelle aggregation. Boirie, *et al*. [10] supported these findings *in vivo*, observing a rapid, transient increase in aminoacidemia after consuming a 13C-Leucine WP meal, compared with a slow, prolonged increase after a casein meal (summarised in Table 2). While the two meals were not iso-nitrogenous, with more protein being present in the whey-based meal (336 mmoles nitrogen in WP meal vs. 479 mmoles nitrogen in casein meal), it was still found that the casein meal produced a lower aminoacidemia peak than WP, where leucine was 236 ± 56% above baseline after 100 min of digestion of the WP meal, compared with just 77 ± 24% in the casein meal. Although gastric emptying rates were not determined, there is a suggestion that the coagulation of the casein-based meal led to a slower gastric emptying. This was the causal factor of a gradual release of amino acids into the blood stream from casein compared with WP.

The different digestion profiles of casein and WP can have physiological implications, with one of the outcomes that has been investigated being the relationship with muscle protein synthesis. Research in rats has shown that the co-consumption of casein and WP, combining the rapid digestion and absorption of WP with the slower kinetics of casein, can increase muscle protein synthesis compared with either WP or casein ingestion alone [72]. In humans, WP alone has been shown to be more effective at stimulating muscle protein accretion than casein alone [73]. The difference in muscle protein synthesis between WP and casein may be due to the slower emptying of casein from the gastric phase, the difference in leucine content between the two or a combination of both. WP has a higher leucine content when compared with casein, which may be a causative factor in the previously described difference in muscle protein synthesis between WP and casein, due to leucine’s activation role in key muscle mass regulation pathways (specifically the mTOR pathway) [74,75].

As well as this, WP has been suggested to increase short term satiety more effectively than other protein sources such as egg albumin and soy protein, as well as other macronutrients, fat and carbohydrates [76,77]. This has been supported by longitudinal studies, whereby the increased consumption of dairy proteins was correlated to lower weight gain [78]. These findings have also been replicated more recently in Ireland, where higher dairy intake was associated with positive body outcomes such as lower body mass index and lower body fat [79]. These results are indicators of the positive health outcomes of consuming dairy products.

### 3.2. Milk Protein Polymorphisms

Bovine milk contains six major proteins encoded by six genes. The casein proteins, αS1-casein, β-casein, αS2-casein and κ-casein are encoded by the genes CSN1S1, CSN2, CSN1S2, and CSN3 genes, respectively, all located within a 250 kb region on chromosome 6, while α-LA is encoded by the gene LAA on bovine chromosome 5, and β-LG is encoded by the gene LGB on bovine chromosome 11. Each of these genes are highly polymorphic, with several single nucleotide polymorphisms leading to amino acid changes in the protein sequence. Milk can be homozygous or heterozygous for protein variants, and Table 3 shows the identified variants in bovine milk.

In the past, studies have focused on genetic variations in milk in relation to their effect on the production of dairy products such as cheese and yogurt, with limited study on their effect on digestion [86,87]. However, with the move towards cellular agriculture and precision fermentation-produced dairy proteins, there has been increased interest in the effects of protein polymorphisms on digestion.

#### 3.2.1. κ-Casein

It is currently accepted that 12 variants of κ-casein exist in bovine milk, with the κ-casein variant phenotype of Irish Holstein Friesians being 1.98% BB, 53.07% AA and 44.95% AB, with the E variant also being present to some extent [88]. When comparing the A and B variants, two substitutions are observed: Thr→le136 and Asp→Ala148. These substitutions replace hydrophilic residues in κ-casein A with hydrophobic residues in κ-casein B [89]. One amino acid substitution seen when comparing A vs. E is a Ser→Gly substitution at AA 176 [90].

Recently, the interest in the effects of κ-casein polymorphism on milk digestion has increased. Petrat-Melin, *et al*. [91] investigated the gastrointestinal digestion of isolated κ-casein variants *in vitro*. As expected with casein proteins, all studied variants (A, B, and E) of κ-casein were highly digestible in the gastric phase, with none remaining intact after gastric digestion. However, different digestion kinetics were observed. It was found that after 20 min of gastric digestion, the degree of hydrolysis of variants A and B was significantly higher than that of the E variant. Although this difference was no longer observed at the end of digestion, the authors suggested that the E variant was less susceptible to digestion due to the Ser155 to Gly155 substitution causing a conformational change to the peptide bond, or due to the increased glycosylation seen in the E variant compared with the others. Furthermore, Sheng, *et al*. [21] carried out *in vitro* digestion of milks containing different κ-casein phenotypes, AA, BB or AB. This study found κ-casein AA to be more rapidly hydrolysed during the gastric phase than other variants, with SDS-PAGE data showing this variant almost completely hydrolysed after five minutes of gastric digestion. Similarly to the study by Petrat-Melin, *et al*. [92] described above, no differences were observed during the intestinal phase. However, these studies did not mimic the dynamics of gastric emptying. This study did not use purified κ-casein variants as was seen in Petrat-Melin, Andersen, Rasmussen, Poulsen, Larsen and Young [92]; therefore, the risk of confounding factors was increased, such as the recorded heterogeneities in the phenotypes of other casein types, especially the occurrence of different β-casein variants. These confounding factors were eliminated in a subsequent study by Sheng, *et al*. [93], whereby purified κ-casein variants were digested using the standardised INFOGEST static *in vitro* digestion protocol. This study was of interest as it incorporated peptide profile analysis post digestion to give an insight into the non-digested peptides and potential bioactive peptide release. The results showed that during gastric digestion, κ-casein B releases significantly less peptides when compared with both A and E, indicating that B is more resistant to gastric digestion.

The observed difference in degree of hydrolysis may be due to differences in post-translational modifications (PTMs) of κ-casein variants. PTMs occur in the Golgi apparatus located in the mammary glands. It has been shown that κ-casein can be phosphorylated at one, two or three sites; however, it is more common for κ-casein to have two phosphorylation sites, while it is less common to have either no phosphorylation or to have all three sites phosphorylated [21,88]. These PTMs are influenced by genetic variants. Despite κ-casein A having more possible O-glycosylation sites (0–6) due to the presence of an extra Thr amino acid compared with κ-casein B (0–5) in the C-terminus, κ-casein B has been shown to be more highly glycosylated than A [94]. The complexity of κ-casein is furthered by the existence of 0–2 phosphorylations due to the presence of Ser residues. Moreover, the N-terminus can participate in intermolecular disulphide bridging via its Cys residues to αS2-casein [95]. Research has shown that phosphorylated peptides may be more resistant to tryptic hydrolysis than non-phosphorylated variants due to the existence of salt bridges in close proximity to cleavage sites [96,97]. However, the cited studies did not use individual milk proteins, nor did they use the standardised INFOGEST *in vitro* digestion method; therefore, the hypothesis that the degree of phosphorylation may affect gastrointestinal digestion requires further investigation.

Another PTM that occurs in κ-casein is glycosylation. Glycosylation occurs exclusively at Thr residues via o-linked glycosylation. Glycosylation increases the ability of casein micelles to repel each other due to increased steric charges. In terms of digestion, it has been shown that the action of the main digestive enzymes involved in protein hydrolysis, trypsin, chymotrypsin and pepsin, are negatively impacted by glycosylation [98]. Interestingly, Boutrou, *et al*. [99] have shown differences in digestion between glycosylated and un-glycosylated caseinomacropeptide (CMP) regions of κ-casein A, whereby decreased numbers of peptides were liberated from glycosylated CMP compared with non-glycosylated, indicating a resistance to digestion provided by O-Glycosylations. However, these CMPs were highly processed in order to be isolated, and this investigation is difficult to replicate due to the use of a freshly slaughtered pig to isolate brush border membrane vesicles. In an *in vitro* gastrointestinal digestion study, it was also shown that the areas containing glycosylation were resistant to hydrolysis in κ-casein variant A; however, no other variants were studied, and the number of glycosylation’s present was not elucidated [100]. The effect of glycosylation on casein micelle size and stability is still debated. Bijl, *et al*. [101] hypothesised that the lower casein micelle size seen in κ-casein AB compared with AA was due to the B variant being more highly glycosylated; however, other studies have shown contradictory results [102,103]. When isolated κ-casein protein variants were subjected to *in vitro* digestion, it was found that the variants with higher glycosylation showed a lower degree of hydrolysis: for example, variant E was highly glycosylated and showed a lower degree of hydrolysis than other variants [91]. Sheng, *et al*. [21] also investigated the degree of glycosylation of different κ-casein phenotypes in skim milk, linking the lower degree of glycosylation in AA milk to smaller particle sizes when compared with milk containing either AB or BB κ-casein. In the same study, Sheng, *et al*. [21] subjected the skim milk containing different κ-casein variants to static *in vitro* digestion, finding that AA was hydrolysed more rapidly in the gastric phase; however, no differences between variants were observed after intestinal digestion. Since the amino acid substitutions that are observed between κ-casein A and B are not potential hydrolysis sites for digestive enzymes, it can be assumed that differences in digestion are brought about by factors other than a change in the number of hydrolysis sites. Differences in hydrophobicity and net negative charge could be the causative factors in differences in digestion. This observation is further supported by the findings of Sheng, *et al*. [93], who utilised peptide analysis to show that the least number of peptides released from any κ-casein region of any variants were in the region of glycosylation sites, indicating that glycosylation inhibits enzymatic activity. Another interesting finding from this study was that while all peptides released from the non-glycosylated part of κ-casein were the same for all variants, no peptides released from the glycosylated part of κ-casein were shared.

The size of the casein micelle may be of importance when studying the effects of κ-casein variation on digestion kinetics due to the observed effects on dairy processes such as renneting. The average diameter of casein micelles in bovine milk is approximately 200 mm [104]. Smaller diameter micelles are optimal for gelation, such as in renneting [105]. In a study by Devold, *et al*. [106], κ-Casein AB variants were associated with smaller micelle diameters when compared with either AA or AE κ-casein variants. This was in agreement with Sheng, *et al*. [21], who found a larger casein micelle size in skim milk containing κ-casein AA compared with AB or BB. On from this, since the concentrations of κ-casein in a milk sample can affect the size of the micelle, it is important to examine polymorphisms in the non-coding regions of the κ-casein gene. The κ-casein to total casein ratio has been found to be of importance to micelle size, and therefore influences casein digestion, and κ-casein variants have been shown to influence the total amount of kappa casein present. Laible, *et al*. [107] showed that by genetically upregulating the production of κ-casein, a decrease in micelle size could be observed, indicating that a higher concentration of κ-casein results in smaller diameter casein micelles. Studies analysing the effect of micelle size on digestion are limited; however, there are some data available on milk samples from various species with different micelle sizes. For example, equine milk contains a low concentration of κ-casein when compared with bovine milk, and since an inverse relationship between κ-casein content and micelle size exists, casein micelles from equine milk are significantly larger than bovine milk, at 311 and 184 nm on average, respectively [108,109]. Having noted this, it has been shown, *in vitro*, that the gastric digestion of equine casein occurs more rapidly than in bovine casein, with a 70% digestion rate after 30 min, compared with 31% in bovine milk [109]. Differences in the *in vitro* digestion of bovine milk and sheep milk have also been found, linked to a decreased concentration of κ-casein found in sheep milk. This may lead to the finding of increased curd firmness of curds formed during gastric digestion in sheep milk, resulting in a less extensive breakdown of proteins [110]. However, when comparing milk of different species, it is difficult to overcome the obstacle of confounding factors; therefore, no assumptions can be made until further investigation is carried out on the effect of micelle diameter on digestion in bovine milk.

The study of the effect of κ-casein variants on digestion kinetics has shown some interesting results, most interestingly, changes to the degree of hydrolysis correlated to different κ-casein variants. To our knowledge, no study has investigated the effect of these variants in a dynamic digestion system, *in vivo* or *in vitro*. Further studies should be carried out to determine differences in gastric emptying rates due to genetic variations of κ-casein using semi-dynamic or fully dynamic *in vitro* digestion systems. This work could be furthered by the use of MRI scanners to analyse the differences caused by κ-casein variants *in vivo*. It is hypothesised that differences in rates of digestion and the degree of hydrolysis rates of nutrient absorption could be affected by κ-casein genetic variants.

#### 3.2.2. β-Casein

Beta-casein (β-casein) is another casein in bovine milk that is of interest in terms of genetic polymorphisms. β-Casein accounts for 40% of the casein fraction of bovine milk and is coded by a highly polymorphic gene, CSN2, located on chromosome 6. Presently, there are 15 known variants of β-casein, with A1, A2 and B most commonly found in bovine milk [111]. The release of bioactive peptides from β-casein is a current topic of interest, and has been of interest since the opioid activity of a β-caseinomorphin (BCM) was first discovered [112]. BCMs are a group of peptides with chain lengths between 4 and 11 amino acids that all begin with Tyr, and they lie dormant within the β-casein protein in milk until released by the action of gastric pepsin during digestion. The BCM first isolated by Brantl, *et al*. [112] was BCM-7, an opioid peptide that is seven amino acids in length (Tyr-Pro-Phe-Pro-Gly-Pro-Ile). An opioid peptide is one that has similar mechanisms of action to morphine, attaching to the same opioid receptors. BCM-7 has continued to be the most highly investigated bioactive peptide to this day due to claims that is has both positive (attenuated aging [113]) and negative impacts (reduction in oxidative stress) [114] on health [115].

Several studies have examined the effect of β-casein genetic polymorphisms on gastrointestinal digestion and health-related outcomes. These studies have been systematically reviewed by Brooke-Taylor, *et al*. [116] and more recently by Daniloski, *et al*. [115], mainly focusing on the release and effect of BCM-7. As well as this, Daniloski, *et al*. [117] has also reviewed *in vitro* and ex vivo studies relating to β-casein and β-caseinomorphins. Daniloski, *et al*. [117] reviewed the potential impact of BCM-7 on human health from ex vivo studies, finding that they may have potential impacts on human health. However, as highlighted in a subsequent review, despite increases in the release of BCM-7 from A2 vs. A1 β-casein, Daniloski, *et al*. [115] did not find evidence of clinically relevant health outcomes associated with β-casein genetic polymorphisms.

#### 3.2.3. β-LG

The digestion of whey proteins such as β-LG is also affected by genetic polymorphisms. The coding of B-LG occurs on chromosome 11 by the gene LCG, with 11 variants of the protein having been discovered, with B, A and C being most common in European dairy herds [118]. Bovine milk can be heterogeneous, in that it produces two variants of β-LG, or it can be homozygous to one. Bewley, *et al*. [119] determined the structural differences between variants using X-ray crystallography, showing high degrees of similarities between the variants. However, local internal readjustments have been noted due to the Gly64Asp (A to B) mutation, as well as a difference in surface stabilization due to the Gln59His mutation (B to C) [119]. Studies examining B-LG variants and their effects on cheese production show significant differences in rennet coagulation time and curd firmness, indicating that there may also be a difference in terms of human digestion [120,121]. Creamer, *et al*. [122] determined the differences in tryptic hydrolysis of β-LG A, B and C using SDS-PAGE and HPLC. This study indicated that the A variant is more susceptible to hydrolysis than the other two variants, and that the relative changes in rates of hydrolysis of B-LG A in response to environmental changes was significantly greater than B and C which indicates that a significant structural difference exists between B-LG A compared with the other two variants. This study also agrees with Bewley, *et al*. [119], suggesting that the Val118Ala (A to B) mutation may play an important role in digestion because it is located near Cys119, which forms a disulphide bond within the FGH motif. The Asp64Gly mutation is also of interest, as the A variant introduces an additional acidic residue in the CD loop, possibly influencing the dimer–monomer dissociation equilibrium [119,122].

Differences in thermal stability of β-LG genetic variants have been observed by O’Loughlin, *et al*. [123], who demonstrated that β-LG B is more resistant to heat treatment than β-LG A. Schmidt and Markwijk [124] also showed that all variants of B-LG are more rapidly hydrolysed by pepsin post-heat treatment compared with native β-LG, with heating having a greater effect on the A variant than the B variant.

Differences also exist in terms of β-LG digestion between species. When bovine milk was digested using *in vitro* digestion, it was found that 83% remains intact at the end of gastric digestion; however, when caprine milk is digested, just 23% of β-LG remained intact [125]. This finding is interesting as the structure of β-LG is highly conserved between these species. Although there are six amino acid substitutions between caprine β-LG and the B variant of bovine β-LG, these substitutions do not appear to result in tertiary or quaternary structural changes [126]. Further, β-LG was found to be digested more extensively *in vitro* in bovine vs. sheep milk and yogurt. This was due to the formation of a denser curd in sheep’s milk, reducing the access of proteases to proteins [110].

## 4. Extrinsic Factors Affecting Milk Protein Digestion

### 4.1. Processing

To ensure the microbiological safety and to increase the shelf life of commercial milk, various heat treatments can be used. According to the International Dairy Foods Association (IDFA), the most common heat treatment consists of heating to 72 °C for 15 s/90–95 °C for 20 min (High Temperature Short Time Pasteurisation, HTST) [127]. Milk can also be heated to 63 °C for 30 min (Vat Pasteurisation, Low Temperature Long Time, LTLT), or to 138 °C for 2 s (Ultra High Heat Treatment, UHT). As alternatives to heat treatment, non-thermal techniques such as microfiltration and ultra-filtration can also be used to decrease the bacterial load of bovine milk. In addition to thermal/non-thermal treatments aimed at controlling the microbiological load, commercial milk typically undergoes a mechanical treatment known as homogenisation. This process can be briefly described as passing heated milk under high pressure through a small orifice, results in reduction of the average diameter of fat globules, and an increase in their total number and surface area [128]. This leads to a reduced tendency for the creaming of fat globules, ultimately ensuring a uniform, safe and high quality product reaches consumers. These processes have been shown to affect the digestion of milk proteins, which will be discussed in the following sections.

#### 4.1.1. Heat Treatment

The heat-induced changes that occur in milk proteins during processing can affect the digestion of both WPs and caseins. HTST and microfiltered milk (which undergoes no heat treatment) have similar effects on postprandial aminoacidemia [129]. In contrast, UHT milk leads to a more rapid increase in postprandial serum amino acids in humans [129]. This may be due to differences in the formation of softer coagulation during digestion in UHT and HTST milk due to decreased casein micelle stability in UHT milk [129]. *In vitro* studies offer further mechanistic insight into the digestive processes of each of these milks. It has been shown, for example, that the association of WPs with casein micelles due to pasteurisation leads to softer coagulum formation during gastric digestion [130]. Ye, *et al*. [131] found a higher amount of β-LG and α-LA in clots formed during the gastric digestion of heated milk (90 °C, 20 min) versus unheated whole milk, leading to softer curds in the heated milk. The authors of this study hypothesised that softer clots in heated milk likely facilitate pepsin to hydrolyse protein within the clot structures, increasing the rate of digestion, whereas accessibility was probably hindered in the denser clots formed in unheated milk explaining the slower digestion. Similarly, in another study using a more complex, fully dynamic model, it was found that heat treatment (90 °C, 10 min) led to a decreased rate of casein hydrolysis, but increased rates of β-LG hydrolysis in skimmed milk compared with unheated skimmed milk [132]. Sánchez-Rivera, Ménard, Recio and Dupont [132] also identified differences in peptide fragments released due to heat treatment, which may have an influence on the bioactive properties of the milk. Increasing heat treatment can further influence digestion profiles. Higher temperatures during heat treatment can lead to increased denaturation of WP, leading to an increase in WPs bound to the casein micelle, reducing the ability of casein micelles to interact with one another [133]. Mulet-Cabero, Mackie, Wilde, Fenelon and Brodkorb [23] observed, using *in vitro* semi-dynamic digestion, that heating milk caused the production of more fragmented particles during gastric digestion. This was more pronounced for pasteurised milk (72 °C, 15 s) than UHT milk (140 °C, 3 s).

Recently, *in vivo* studies have used non-invasive techniques such as Magnetic Resonance Imaging (MRI) to study the effects of heat treatment on milk digestion in humans, with conflicting results. Milan, *et al*. [134] found a greater gastric content volume throughout digestion when comparing UHT milk with pasteurised milk, indicating that UHT milk was emptied to the intestinal phase slower than pasteurised milk. This finding contradicts both *in vivo* findings and *in vitro* findings. Mayar, *et al*. [135] used MRI to show that no difference in gastric coagulation is associated with higher heat treatment (90% WP denaturation) compared with lower heat treatment (3% WP denaturation); however, higher heat treatment did result in faster gastric emptying times than lower heat treatment. Although Milan, *et al*. [134] observed findings that seem to contradict *in vitro* digestion studies, which suggest that increasing the heat treatment of milk results in the formation of weaker curds that are emptied more rapidly from the stomach [23], the results of Mayar, *et al*. [135] agrees with these *in vitro* assumptions. When conflicting results such as these are obtained, it can be useful to return to early studies published on the digestion of milk proteins. As discussed in Section 2, after 3 h of digestion of boiled milk, patients stomachs had emptied, while raw milk remained, suggesting a slower rate of gastric emptying in raw vs. heat-treated milk [17]. While Mayar, *et al*. [135] did not use raw milk, they did show that lower heat treatment resulted in slower gastric emptying rates, in agreement with early studies. The advancement in MRI techniques should significantly advance our understanding of factors affecting milk protein digestion *in vivo*.

Heat treatment may also result in the occurrence of Maillard reactions, which may reduce the bioavailability of certain amino acids. For example, if reducing sugars such as lactose bind to Lys residues at the trypsin cleavage site, Lys may become bio-unavailable [136].

#### 4.1.2. Homogenisation

The impact of milk homogenisation on digestion has long been a subject of study. Important learnings can be derived from research on infant nutrition. As early as 1915, different researchers were interested in applying homogenisation to milk, hypothesising that the “more finely divided the food, the greater its accessibility to digestive fluids, and the greater its assimilation” [137]. For example, it has been observed that the homogenisation of human milk significantly increased gastric lipolysis, enhanced the proteolysis of serum albumin and reduced the gastric emptying rate in preterm infants [138]. Furthermore, homogenisation was linked to improved absorption of human milk fat in very low-birth-weight infants [139]. Theoretically, in adults, the digestion of bovine milk could also be expected to be impacted by homogenisation; however, studies where homogenisation and heating are investigated separately are lacking. One study comparing the digestion of homogenised pasteurised milk with raw milk found that while there were no differences in postprandial levels of plasma lipids, the concentrations of some fatty acids were significantly higher following the consumption of the homogenised and heated milk [140]. Interestingly, in an *in vivo* study using pigs as human models, it was found that both pasteurisation and homogenisation increased the rate of milk protein hydrolysis; however, only homogenisation increased the rate of amino acids entering the small intestine [15]. *In vitro* studies further elucidate the impact of homogenisation on digestion. Homogenisation has been shown to affect the coagulation properties of milk during *in vitro* gastric digestion. After 10 min of digestion, both non-homogenised and homogenised whole milk and milk form a coagulum; however, the coagulum formed by homogenised milk is more porous. This porous structure may facilitate pepsin’s ability to access proteins, leading to a more extensive breakdown of coagulum, as demonstrated by Tunick, *et al*. [141].

### 4.2. Consumption Temperature

Bovine milk can be consumed at various temperatures, and several human studies have investigated how these temperatures impact the digestive environment. Using coffee as an example, it was shown that consuming different temperature liquids can affect the temperature of the stomach, cooling or warming depending on the temperature of the liquid [142,143]. It is clear that consuming liquids hot or cold can affect the temperature of the stomach for up to 30 min [142,143]. This may affect the activity of pepsin, leading to changes in digestion.

When consuming bovine milk, deviations from body temperature (37 °C) during gastric digestion may impact milk protein digestion kinetics due to changes in pepsin activity at different temperatures [144]. *In vivo* studies with calves showed that cold milk empties more rapidly than warm milk [145]. This may be due to slower coagulation at colder temperatures, causing milk to remain liquid longer, and liquids emptying more rapidly from the stomach than solids or semi-solids [146]. This difference occurs despite a high gastric pH at the beginning of gastric digestion when temperature differences occur due to the optimum activity of pepsin on κ-casein occurring at pH~6 [147]. This allows a rapid decrease in electrostatic repulsion on the surface of casein micelles due to the release of glycomacropeptide from κ-casein despite alkaline conditions [144,147,148]. Yang, *et al*. [149] used a human gastric simulator model to digest skim milk at 4 °C, 37 °C and 50 °C, reporting that cold milk coagulated more slowly than milk at other temperatures and produced softer curds, leading to increased protein hydrolysis at 4 °C due to better pepsin access.

Fitzpatrick, *et al*. [63] used a modified *in vitro* semi-dynamic gastric digestion under MRI to visualise the differences in the digestion of cold (4 °C), body temperature (37 °C) and hot milk (60 °C). Both Yang, *et al*. [149] and Fitzpatrick, *et al*. [63] attempted to replicate *in vivo* gastric temperatures post-consumption of cold or hot beverages. However, Yang, *et al*. [149] began gastric digestion at 4, 37 or 50 °C, i.e., the temperature of the beverages under study, while Fitzpatrick, *et al*. [63] simulated the consumption of milk cooled/heated to 4, 37 or 60 °C but then adjusted digestive conditions for temperature changes observed during ingestion. As a result, in that study, gastric digestion was initiated at 22 °C for cold milk and at 44 °C for hot milk and then the respective rates of return to 37 °C were simulated based on *in vivo* observations.

Despite the reduced temperature extremes in Fitzpatrick, *et al*. [63], similar results were observed, with cold milk showing slower coagulation than milk at 37 °C or hot milk. Differences in proteolysis were found by Yang, *et al*. [149] but not by Fitzpatrick, *et al*. [63], likely due to greater temperature differences and more targeted protein hydrolysis measurements, such as examining the specific hydrolysis of κ-casein, in Yang, *et al*. [149]. These findings show the need to carefully monitor the temperature at which milk is consumed, or simulated to be consumed, during *in vivo* or *in vitro* digestion experiments.

## 5. Conclusions

Bovine milk is a highly digestible source of high-quality proteins. Several factors can influence the rate and extent of protein breakdown during digestion, potentially leading to changes in the rate of uptake of amino acids into the bloodstream which, in turn, can have physiological implications. This review focused on the impact of variations in the profiles of native milk proteins and of the most common processes to which milk is exposed prior to consumption. Certainly, extrinsic factors applied to milk such as heat treatment and homogenisation lead to differences in the digestion of milk. Moreover, more subtle changes are induced by intrinsic factors such as protein polymorphisms, which can affect the nutritional quality of bovine milk, with either potential beneficial or harmful implications on human health. These subtle changes to bovine milk could be considered for the production of premium milk products aimed at certain population cohorts, such as milk with increased casein content to increase the retention time of proteins, prolonging satiety and negating peaks in aminoacidemia. This paper therefore provides both academia and industry with a base of knowledge to further the dairy industry with respect to human health and nutrition.

## Figures and Tables

**Figure 1 foods-13-03683-f001:**
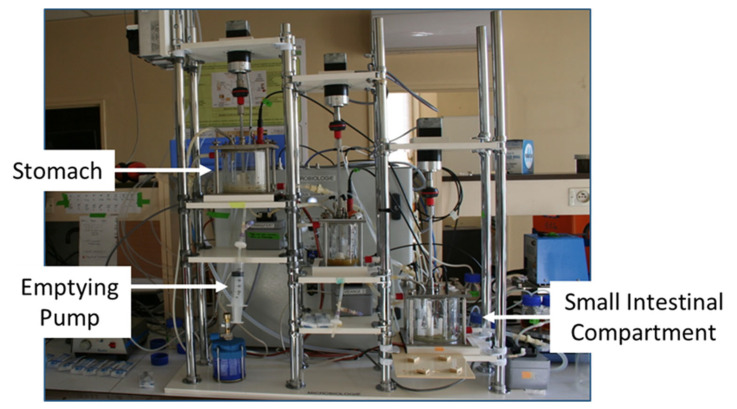
The DIDGI *in vitro* digestion system, STLO, INRAE, France. This is a multi-compartmental system, consisting of both gastric and small intestinal compartments. Food is automatically emptied from one stage of digestion to the next using emptying pumps [12].

**Table 1 foods-13-03683-t001:** Examples of *in vitro* digestion methods that can be used to simulate milk protein digestion in humans. The dynamic system discussed here is one example from several systems available, with other systems discussed in Section 2.

Type	Examples	Characteristics for Milk Protein Digestion
Static	INFOGEST Static model [11]	- Simplified digestion model with fixed enzyme ratios and pH conditions.
		- Does not account for gradual changes in gastric emptying, enzyme secretion or pH.
		- Suitable for comparing protein digestion across different milk formulations.
Semi-Dynamic	INFOGEST Semi-dynamic model [37]	- Simulates dynamic aspects of gastric digestion (e.g., flow of secretions, gradual acidification): milk proteins are exposed to varying pH and enzyme concentrations over time.
		- Simulates gradual changes in gastric emptying, allowing for more realistic protein hydrolysis profiles.
		- Balances simplicity and physiological relevance for digestion studies.
Dynamic	DIDGI Model [12]	- In addition to dynamic conditions mimicked by semi-dynamic systems, dynamic systems allow replication of other dynamic aspects such as continuous gastric emptying. DIDGI system also incoportates a dynamic intestinal phase, mimicking intestinal pH, peristalisis and transit times.
		- Release of FAA closer to *in vivo* situation than INFOGEST static protocol [38].
		- Provides more physiologically relevant data but requires advanced equipment and is time-consuming.

**Table 2 foods-13-03683-t002:** Summary of factors affecting the digestion of milk proteins.

Factors	Description	Outcome for Protein Digestion	Type of Digestion	References
Whey: Casein Ratio	The ratio of whey to casein proteins in milk varies due to factors such as stage of lactation and breed.	Increasing casein content decreases the rate of protein hydrolysis and gastric emptying.	*In vivo* [10] *In vitro* [71]	[10,71]
Protein Polymorphisms	Variants in milk protein genes, such as κ-CN A, B and E, affect protein conformation.	Polymorphisms affect glycosylation levels and casein micelle size, influencing rate of digestion of milk protiens in gastric phase (AA digested more rapidly than BB)	*In vitro*	[21]
Heat Treatment	Denaturation of proteins due to thermal processing, affecting the structure of whey proteins, particularly β-LG, leading to interactions with κ-CN on the casein micelle.	Heat treatment affects coagulum consistency, with increasing heat treatment associated with less firm coagulum, leading to a more rapid protein hydrolysis.	*In vitro*	[23]
Homogenisation	Mechanical process that breaks down fat globules, reducing fat particle size, increasing the surface area for interaction with proteins.	Homogenisation increases the rate of amino acids entering the small intestine from the stomach.	*In vivo*	[72]
Consumption Temperature	Milk consumed at different temperatures affects how quickly it coagulates and digests in the stomach.	Hot milk coagulates faster in the stomach, potentially leading to different gastric emptying rates.	*In vitro*	[64]

**Table 3 foods-13-03683-t003:** Milk proteins and their variants.

Gene	Protein	Variants	References
*CSN1S1*	α_S1_-	A, B, C, D, E, F, G, H, I, J, K, L, M, N, O, P, H	[80,81]
*CSN2*	β-casein	A1, A2, A3, B, C, D, E, F, G, H1, H2, I	[80,82]
*CSN1S2*	α_S2_-	A, B, C, D	[80,83]
*CSN3*	κ-casein	A, B, B2, C, E, F1, F2, G1, G2, H, I, J	[80,84]
*LAA*	α-LA	A, B, C	[80,85]
*LGB*	β-LG	A, B, C, D, E, F, G, H, I, J, W	[80,86]

## Data Availability

The original contributions presented in the study are included in the article, further inquiries can be directed to the corresponding author.

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
