# Peer review of "Variations in Bovine Milk Proteins and Processing Conditions and Their Effect on Protein Digestibility in Humans: A Review of In Vivo and In Vitro Studies"

_foods, 2024, doi:10.3390/foods13223683_

Round 1
Reviewer 1 Report
Comments and Suggestions for Authors
foods-3288300-peer-review-v1
1. The structure and logic of this review should be reorganized. Introduction, the novelty of the work (and how it is filling the current gap) is missing. Has any similar study been published before? What difference does your work make?
2. The observations and explanation of milk protein digestion can be illustrated in a separate section. Furthermore, the main findings and conclusion on the progress of protein digestion can be summarized in a table.
3. What are the main characteristics of in vitro digestion models for protein digestion? Please summarize this information in a table.
4. How to define and characterize the extent and rate of protein digestion during the in vitro and in vivo digestion process?
5. The main findings on the digestion of native milk protein influenced by the variations and processing conditions can be summarized in a table.
6. What are the differences between the in vitro and in vivo studies on milk protein digestion? Please illustrate it.
7. Are there other factors influencing the protein digestion? For example, lipid content, the protein aggregation?
Reviewer 2 Report
Comments and Suggestions for Authors
The manuscript seems sound. However, it needs to be revised.
The manuscript entitled “Variations in bovine milk proteins and processing conditions and their effect on protein digestibility in humans: a review of in vivo and in vitro studies” contains an important study. I have some comments to revise the manuscript.
After studying the article, I have major comments:
Line 57-60: Not understood. Please rephrase it.
Line 111: Please rephrase the sentence.
Line 113: due to the addition of HCl?
Line 116: As well as this, it was found that raw milk coagulates within 10 minutes, while boiled milk coagulated within 8-15 minutes. The tenses are different for verb “coagulate”. Please check.
Line 128: Advanced…
Line 152: How the calcium could be removed from the casein micelles?
Line 166: Dupont et al [….]?
Line 174: The sentence could be improved as digestion word is repeated?
Line 192: suggested…….
Line 277: in between infant and………..
Line 440: Table…..
Line 657: currently in use by the food……..?? The sentence seems incomplete?
Line 668: Why the term “globally” has been used?
Line 728: Please revise the sentences and correct the errors.
Line 740: Please correct the sentence.
Overall comments:
- Please check the grammatical errors throughout the manuscript.
- The manuscript can be made more concise.
English needs to be more concise.
Round 2
Reviewer 1 Report
Comments and Suggestions for Authors
The manuscript has improved significantly to justify a publication.